# Modeling the impact of single-cell stochasticity and size control on the population growth rate in asymmetrically dividing cells

Felix Barber[1,2], Jiseon Min[1], Andrew W. Murray[1,3], Ariel Amir[4]*

**1** Department of Molecular and Cellular Biology, Harvard University, Cambridge, Massachusetts, United States of America, **2** Center for Genomics and Systems Biology, Department of Biology, New York University, New York, New York, United States of America, **3** FAS Center for Systems Biology, Harvard University, Cambridge, Massachusetts, United States of America, **4** John A. Paulson School of Engineering and Applied Sciences, Harvard University, Cambridge, Massachusetts, United States of America

* arielamir@seas.harvard.edu

**Data Availability Statement:** All relevant code required to perform the associated simulations is

## Abstract

Microbial populations show striking diversity in cell growth morphology and lifecycle; however, our understanding of how these factors influence the growth rate of cell populations remains limited. We use theory and simulations to predict the impact of asymmetric cell division, cell size regulation and single-cell stochasticity on the population growth rate. Our model predicts that coarse-grained noise in the single-cell growth rate λ decreases the population growth rate, as previously seen for symmetrically dividing cells. However, for a given noise in λ we find that dividing asymmetrically can enhance the population growth rate for cells with strong size control (between a "sizer" and an "adder"). To reconcile this finding with the abundance of symmetrically dividing organisms in nature, we propose that additional constraints on cell growth and division must be present which are not included in our model, and we explore the effects of selected extensions thereof. Further, we find that within our model, epigenetically inherited generation times may arise due to size control in asymmetrically dividing cells, providing a possible explanation for recent experimental observations in budding yeast. Taken together, our findings provide insight into the complex effects generated by non-canonical growth morphologies.

## Author summary

How rapidly a population of single-celled organisms can grow will strongly impact their long-term success. Prior work has shown that many factors impact this population growth rate, including the rate at which single cells grow, random variability between cells, and whether cells regulate their own size. Here we show that cell division asymmetry can also have a strong impact on the population growth rate. We use theory and computer simulations to study the growth rate of cells that divide asymmetrically, producing one smaller cell and one larger cell with each cell division event. We show that variability in how fast

available at https://github.com/felixbarber/division_asymmetry_growth_rate_simulations.git.

**Funding:** F. B. was supported by the William Georgetti Trust, a Harvard Graduate Merit Award and a Harvard Quantitative Biology Initiative Student Award while conducting this research. A. W. M. thanks NIH grant RO1-GM43987 and the NSF-Simons Center for Mathematical and Statistical Analysis of Biology at Harvard (NSF #1764269, Simons #594596) for support. A. A. acknowledges the support of the NSF CAREER award number 1752024 and the support of the Volkswagen Foundation. The funders had no role in study design, data collection and analysis, decision to publish, or preparation of the manuscript.

**Competing interests:** The authors have declared that no competing interests exist.

single cells grow will still decrease the population growth rate, when asymmetry is moderate or size control is weak, but that cells with strong size control can diminish this decrease by dividing more asymmetrically. We also demonstrate that cell cycle lengths can be *positively* correlated for closely related cells when they both divide asymmetrically and regulate their size. This counter-intuitive result contrasts with previous findings based on cell size regulation in symmetrically dividing cells that if cells grow for "too long" in one cell cycle, this will be corrected for by reduced growth during a shorter, subsequent cell cycle.

## 1 Introduction

Recent years have expanded our understanding of heterogeneity at the single cell level, with clonal populations displaying variability in a range of physiological parameters, including cell generation times (the time between cell birth and division), cell size and gene expression [1–5]. This revolution in single-cell microbiology drove a renewed interest in the effect of heterogeneity on cell fitness, taken here to be described by the exponential population growth rate [6–9]. In contrast, a relatively unexplored factor affecting cell fitness is cell growth morphology; microbial cells display an astonishing degree of variability in growth morphology and life cycle, ranging from symmetric division in the vegetative growth of bacteria such as *Bacillus subtilis* and *Escherichia coli*, to the asymmetrically dividing, budding yeast *Saccharomyces cerevisiae*, to more diverse growth morphologies such as those observed recently in a range of marine yeasts [10]. However, our understanding of the physiological effect of division asymmetry on the population growth rate remains limited.

Early work demonstrated that the population growth rate $\Lambda_P$ obeys the Euler-Lotka equation [11]

$$1 = 2 \int_0^\infty \exp\left[-\Lambda_p t\right] f_0(t) \mathrm{d}t. \tag{1}$$

Here $f_0(t)$ is the distribution of generation times measured by tracking all cells in a growing population (called the lineage tree or tree distribution here), illustrated in Fig 1A [6, 12]. If generation times are uncorrelated between related cells (the independent generation time or IGT case), the tree distribution $f_0(t)$ is equal to the distribution obtained from tracking cells along a single cell lineage. However, this simplification does not hold in the case of correlated generation times, which have been observed in a range of organisms [4, 5, 13–15], meaning the full tree distribution is required for Eq 1 to hold. These generation time correlations are expected as a direct consequence of size control, whereby cells couple their growth and division to constrain the spread of sizes observed throughout a population [16]. The effects of generation time correlations from size control can be substantial; including size control when modeling cell cycle progression fundamentally changes the predicted impact of stochasticity at the single-cell level on $\Lambda_P$ [6]. Prior studies that did not incorporate cell size control have concluded that noise in generation times can enhance the population growth rate [7, 13]. In contrast, studies incorporating cell size control predict that the single cell exponential growth rate $\lambda$ sets $\Lambda_P$, with $\Lambda_P = \lambda$ exactly in the absence of noise in their single cell growth rate [6]. This can be readily shown by requiring that the cell size distribution reaches steady state with a constant average size $\langle V \rangle$, since $\langle V \rangle(t) = \Sigma_i V_i(t)/N(t) \propto \exp[(\lambda - \Lambda_P)t] = constant$, where $V_i$ is the volume of each cell $i$ in the population and $N(t)$ is the population number at time $t$. Coarse-grained noise in $\lambda$ then decreases $\Lambda_P$ below the average single cell growth rate, while for a

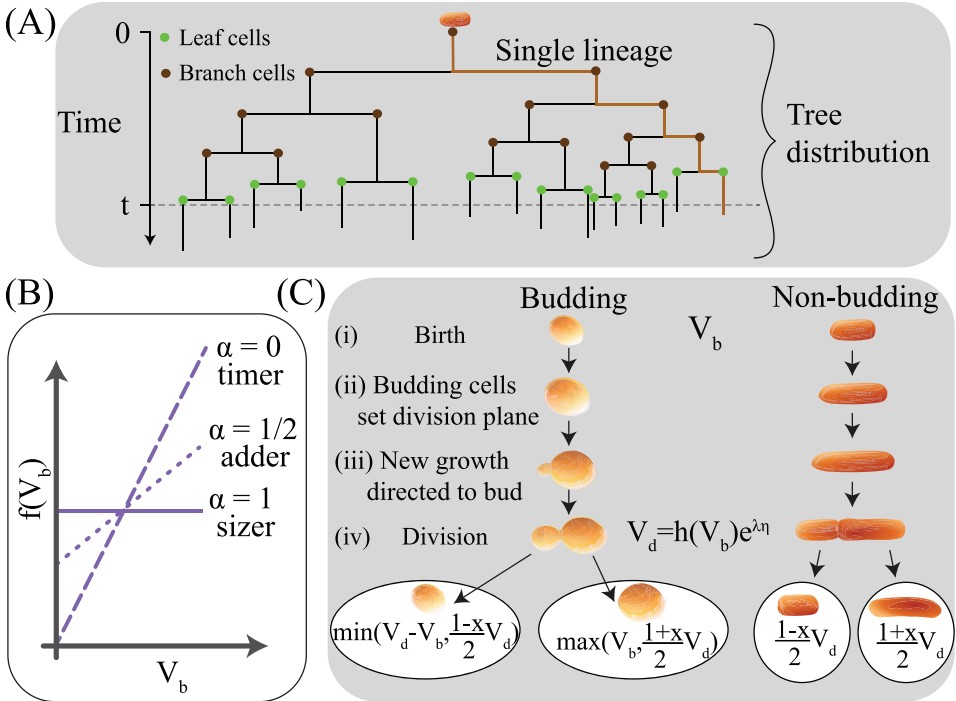

**Fig 1.** (A) Illustration of the tree distribution for a growing population of cells, terminated at a time-point *t*. We note that the tree distribution includes not just the generation times of "branch" cells whose cell cycles have finished at *t*, but also those "leaf" cells that are born before *t* but will complete their cell cycle after time *t*. A single lineage is shown in orange. Each node corresponds to a single cell division event. (B) Illustration of differences in cell size control policy. (C) Illustration of asymmetric division in different growth morphologies. Budding cells set the plane of division early on in the cell cycle and direct growth to a newly forming bud beyond that division plane. In contrast, non-budding cells set the plane of division when division occurs, meaning that growth throughout the cell cycle affects both progeny. The min and max conditions for budding cell size at birth are related to the observation that in a budding morphology, mother cells will only ever increase in size over subsequent cell cycles (see main text for details). In the expression for volume at division $V_d$, $\eta$ represents time-additive noise in the timing of cell division.

given noise in λ, increasing noise in generation times is predicted to only have a smaller, secondary effect [6, 8].

Asymmetric cell division generates two distinct cell types; in budding yeast these are known as daughters (the smaller cells) and mothers (the larger cells). To compensate for this difference in size, daughters have a longer average generation time than mothers. One early study in budding yeast theoretically predicted the dependence of $\Lambda_P$ on the division times for daughters ($\tau_D$) and for mothers ($\tau_M$), with $\tau_M$ and $\tau_D$ assumed to be constant [14] (Section 3 in S1 Appendix). A more recent study computationally explored the effect of correlated generation time noise on the population growth rate of budding yeast cells [13]. However, as discussed above, this work did not employ a model of cell size control, leading the authors to predict that single cell stochasticity and epigenetically inherited generation times can enhance the population growth rate. Our results disagree with these predictions. Here we show that for cells that regulate their size, the population growth rate is set primarily by the single cell growth rate λ, with noise in λ *decreasing* the population growth rate, as in the case of symmetrically dividing cells. We further show that asymmetric division can *increase* the population growth rate, and that epigenetically inherited generation times can arise as a natural consequence of size control in asymmetrically dividing cells.

## 2 Results

### 2.1 Model for asymmetric population growth

As discussed above, when the single cell growth rate $\lambda$ is constant, $\Lambda_P = \lambda$ exactly [6]. To study the effect of finite noise in $\lambda$, we modeled the growth of two coupled cell populations ($N_M$ for mothers and $N_D$ for daughters). The growth of these populations is described in the limit of large population numbers by

$$\frac{\mathrm{d}N_D}{\mathrm{d}t} = \Lambda_M N_M(t),$$
$$\frac{\mathrm{d}N_M}{\mathrm{d}t} = \Lambda_D N_D(t), \tag{2}$$

since a cell of either type divides to give one new cell of each type. Here $\Lambda_D$ and $\Lambda_M$ each correspond to the division rate per cell of type $D$ and $M$ respectively. Assuming steady state composition of the population, with a constant relative difference in the number of different cell types $m(t) \equiv (N_D(t) - N_M(t))/(N_D(t) + N_M(t)) = m$ (which we will corroborate later), the full population $N(t) = N_D(t) + N_M(t)$ will grow exponentially with growth rate $\Lambda_P = \sqrt{\Lambda_D \Lambda_M}$ (Section 1 in S1 Appendix). Importantly, the Euler-Lotka equation for the two population system described above still holds (Section 2 in S1 Appendix):

$$1 = 2 \int_0^\infty \exp\left(-\Lambda_P \tau\right) f_0^P(\tau) \mathrm{d}\tau. \tag{3}$$

Here $f_0^P(\tau) = \frac{1}{2}\left(f_0^D(\tau) + f_0^M(\tau)\right)$ is the distribution of generation times measured over the full lineage tree, including both mother and daughter cells. A corresponding constraint equation also exists for the relative difference in population numbers $m$ (Equation 11 in S1 Appendix). We note that although $m$ will in general be greater than zero, with a larger fraction of daughter cells than mother cells at a given point in time, the populations of daughter and mother cells will be equal in size when measured over the full lineage tree, leading to the factor of 1/2 in the definition of $f_0^P(\tau)$. In the case with no generation time correlations, Eq 3 can be derived simply in a similar manner to that described in [6] by noting that a population seeded from a single cell that divides at time 0 will grow exponentially as $e^{\Lambda_P t}$, which must match the size of the two populations seeded by the two progeny cells, with combined sizes growing as $2 \int f_0^P(\tau) e^{\Lambda_P(t-\tau)} \mathrm{d}\tau$. Dividing through by $e^{\Lambda_P t}$ then yields the desired result. Section 2 in S1 Appendix details our derivation of Eq 3 for the case of correlated generation times, similar to a prior approach that addressed symmetrically dividing cells [6]. Eq 3 can also be shown to apply in the case of finite population sizes using the transport equation approach outlined in Ref. [17]. Our current approach provides the additional benefit of predicting the ratio of cell type population sizes $m$ present in the population at a single time-point, given the distribution of interdivision times.

### 2.2 Models of size control

To study the effect of size control on $\Lambda_P$, we define a growth function $h(V_b)$ that sets the target volume at division to be a linear function of volume at birth $V_b$, with a tunable parameter $\alpha$ [18]:

$$h(V_b) = 2\alpha\Delta + 2(1 - \alpha)V_b. \tag{4}$$

Setting $\alpha = 0$ gives a timer model, in which cells grow to double their volume between birth

and division, while $\alpha = 1/2$ gives an adder model with a constant volume $\Delta$ added between birth and division, and $\alpha = 1$ gives a sizer model where cells grow to a threshold size $2\Delta$ at division (see Fig 1B). Prior work has shown that an adder model with $\alpha = 1/2$ effectively captures the size-dependence of cell cycle progression in diploid, daughter budding yeast cells, indicating that Eq 4 is adequate as a generic model of cell size control [5]. Cell volume at division is then given by $V_d = h(V_b)\exp[\lambda\eta]$, with associated generation time $t = \ln|h(V_b)/V_b|/\lambda + \eta$ where $\eta \sim \mathcal{N}(0, \sigma_t^2)$ is a coarse grained noise in generation times (independent and identically distributed, I.I.D., for each newborn cell) and $\lambda$ is the I.I.D. exponential single cell growth rate taken to be $\lambda \sim \mathcal{N}(\lambda_0, \sigma_\lambda^2)$. We define the parameter $x$ as the relative difference in volume at birth between the daughter and mother cells produced from a given division event: $x = (V_b^M - V_b^D)/(V_b^M + V_b^D)$ [19], as described in Fig 1C. This implies $0 < x < 1$. We will use subscripts $b$ and $d$ to denote whether the cell volume is evaluated at birth or at division, while the superscripts $D$ and $M$ correspond to the two different cell types. When a statement is independent of cell type we use the superscript $P$ to denote that cell. Our prior work has studied the differences between budded cells and non-budded cells as shown in Fig 1C [20]. The most prevalent example of an asymmetrically dividing, budding cell is *S. cerevisiae*, while asymmetrically dividing, non-budding cells include the Gram-negative bacterium *Caulobacter crescentus*, as well as various asymmetrically dividing mycobacteria [14, 21, 22]. Budding cells establish the plane of division early in the cell cycle at the boundary of the cell, and from then on direct further growth towards a newly forming bud on the other side of the division plane. This has the consequence that the volume of mother cells will only ever increase when tracked over multiple generations, whereas for non-budding cells, a mother cell may be born smaller than its parent cell was at birth. As mother cells grow progressively larger, the relative amount of growth added with each new cell cycle can become too small to allow the growing bud to reach the desired ratio of daughter cell size to mother cell size. To address this case, and to ensure that cell size only increases with each generation in our simulations, our model for a budding morphology applies the condition for mother cells that $V_b^{n+1} = max(V_b^n, (1 + x)V_d^n/2)$. In this case we assume that the division timing still follows Eq 4. This "maximum" condition implicitly assumes that these large cells will begin budding immediately upon birth, producing daughter cells that represent a smaller fraction of the mother cell size than would be expected based on the division asymmetry we typically impose. In fact, without time-additive noise this "maximum" condition for budding cells is never invoked, since $max(V_b^n, (1 + x)V_d^n/2)$ consistently favors the second term. This can be readily seen, since by following the growth policy of Eq 4 with a given division asymmetry $x$, a mother cell's size at birth will increase monotonically over successive generations (i.e. $V_b^{n+1} = (1 + x)V_d^n/2 > V_b^n$) towards a fixed point at $V_b^* = \alpha\Delta(1 + x)/(\alpha(1 + x) - x)$. Cells born at this fixed point will begin budding immediately, and will produce daughter cells whose size is consistent with the division asymmetry we impose. A sizer model without time-additive noise represents a special case of this, wherein all mother cells will begin budding immediately. Since the "maximum" condition is never invoked when time additive noise is zero, budding cells will grow identically to non-budding cells in this case, as demonstrated in S1(A) Fig for $\sigma_t = 0$. Consequently, within our implementation of budding, our analytical results for the population growth rate without noise in generation times are consistent for both budding and non-budding morphologies. In the presence of generation time noise this equality need not hold; however, the effect of growth morphology on the population growth rate remains minimal in this case as seen in S1(A) Fig for $\sigma_t > 0$. We therefore concluded that the choice of budding vs. non-budding growth morphologies did not impact the population growth rate substantially, and all results presented herein are valid for both budding and non-budding morphologies unless explicitly stated otherwise. For our

investigations of generation time correlations, the general conclusions are also independent of growth morphology, but for completeness we present simulated results for both growth morphologies to highlight the relevant differences. We now study the effects of variation in division asymmetry $x$, size control strategy $\alpha$ and noise terms $\sigma_\lambda$ and $\sigma_t$ on $\Lambda_P$.

## 2.3 An approximate solution for the population growth rate

We first tested the prediction that $\Lambda_P = \lambda$ exactly when $\sigma_\lambda = 0$ by simulating the growth of populations of cells, using Eq 4 to regulate the timing of division (see Methods for details). Fig 2A demonstrates that $\Lambda_P$ is indeed independent of $\sigma_t$ and $x$ for $\sigma_\lambda = 0$. This result is plotted only for the case of adder cells ($\alpha = 1/2$), but holds for any strategy of size control. We further observed that if $\sigma_\lambda$ is nonzero, the population growth rate then decreases further below $\langle \lambda \rangle$, consistent with behavior seen previously in symmetrically dividing cells [6], while in contrast, noise in generation times $\sigma_t$ has a small, secondary effect in the range of biologically relevant division asymmetry values. The negative impact of noise in the single cell growth rate on the population growth rate is on the order of 1.5% for the biologically relevant case of $\sigma_\lambda/\lambda_0 \approx 0.15$ [4], indicating that this effect may be significant from an evolutionary standpoint. We note that the effect of $\sigma_t$ becomes more substantial in the regime of extreme division asymmetry, as shown in S1(A) Fig, however, this regime is not believed to be biologically relevant based on experimental measurements of the division asymmetry $x$ in budding yeast ranging from 0.2 to 0.35 across different growth conditions [5].

We studied the dependence of the population growth rate $\Lambda_P$ on division asymmetry for finite $\sigma_\lambda$, both computationally and analytically. Solving Eq 3 to infer the population growth rate for a general size regulation model is difficult due to correlations between successive generation times. These correlations vanish for symmetrically dividing cells without noise in generation times that follow any mode of size control, as has been studied previously [6]. To gain traction on this problem, we therefore exploited the fact that these correlations also vanish for asymmetrically dividing cells following a sizer model ($\alpha = 1$) without time-additive generation time noise. By applying a saddle point approximation to Eq 3 and Taylor expanding in $\Lambda_P$ and $\sigma_\lambda$ we obtained an approximate solution valid for a sizer model with $\sigma_t = 0$ (see Section 3 in S1 Appendix for details):

$$\Lambda_P(\sigma_\lambda, x) = \lambda_0 \left( 1 - \left( 1 + \frac{1}{2} \frac{(1+x)\left(\ln\left(\frac{1+x}{2}\right)\right)^2 + (1-x)\left(\ln\left(\frac{1-x}{2}\right)\right)^2}{(1+x)\ln\left(\frac{1+x}{2}\right) + (1-x)\ln\left(\frac{1-x}{2}\right)} \right) \left(\frac{\sigma_\lambda}{\lambda_0}\right)^2 \right) + O(\sigma_\lambda^4). \quad (5)$$

From Eq 5, we predict that noise in $\lambda$ will tend to decrease the population growth rate as in the case of symmetric division [6]. However, we see that Eq 5 further predicts that for non-zero $\sigma_\lambda$, $\Lambda_P$ can be increased by increasing the division asymmetry $x$. We tested Eq 5 for a sizer model against simulations across a range of values for $\sigma_\lambda$ and $x$ (with $\sigma_t = 0$), finding consistently good agreement as shown in Fig 2B and S1(B) Fig. These findings make the strong prediction that increasing division asymmetry can enhance $\Lambda_P$ for cells that regulate their size with a sizer strategy and have non-vanishing growth rate variability. We further note that setting $x = 0$ recovers the approximate solution for symmetric growth [6], with

$$\Lambda_P(\sigma_\lambda) \approx \lambda_o \left( 1 - \left( 1 - \frac{\ln 2}{2} \right) \left( \frac{\sigma_\lambda}{\lambda_o} \right)^2 \right). \quad (6)$$

To explore whether increasing division asymmetry consistently increased $\Lambda_P$ for different strategies of cell size control within our model, we simulated population growth across a range of $\alpha$ values between 0 and 1. Results are plotted in Fig 2C, showing that for cells that divide

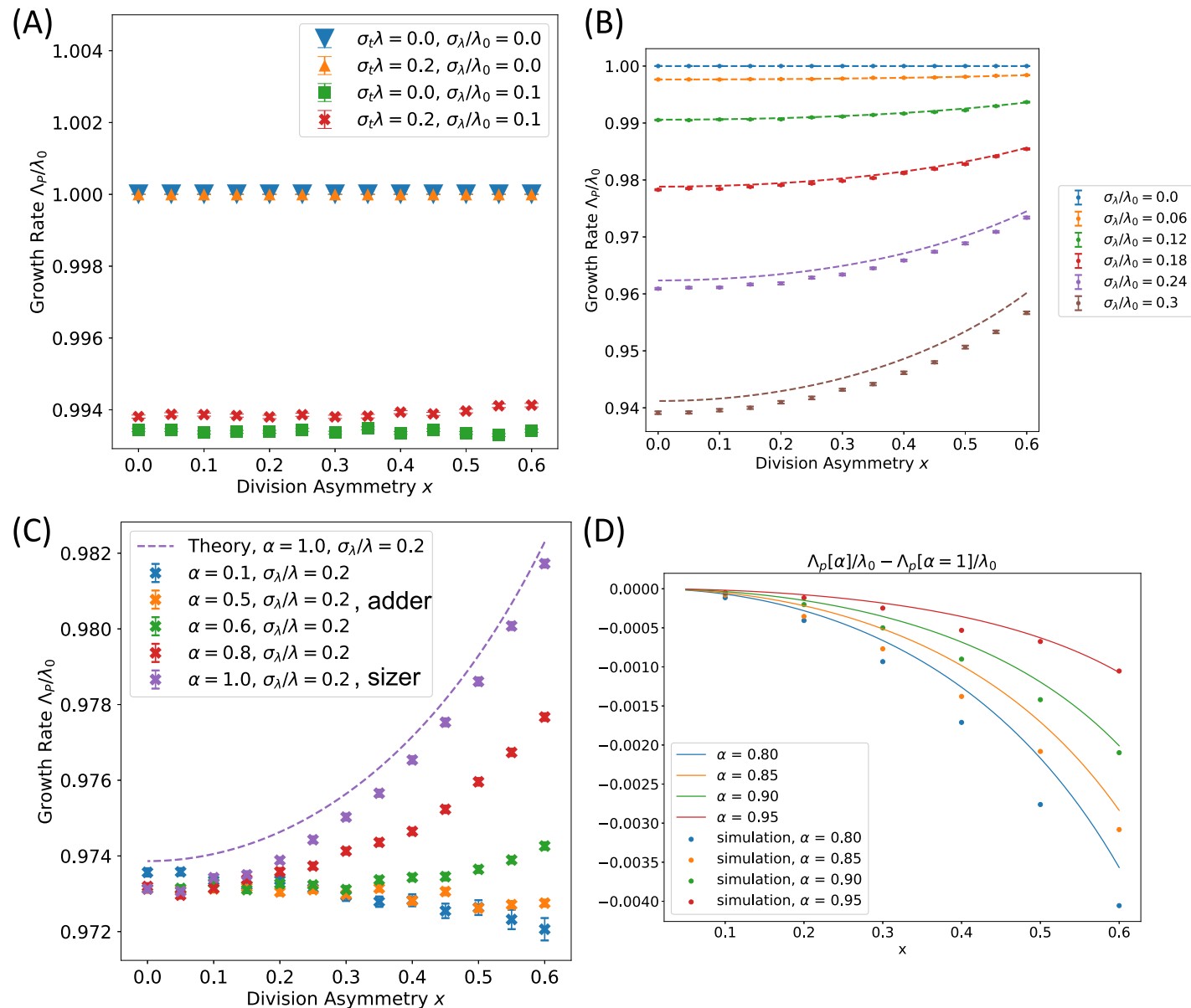

**Fig 2. The population growth rate $\Lambda_P$ is dependent on noise in single cell growth rate $\sigma_\lambda$, the division asymmetry $x$ and the size control strategy $\alpha$ for asymmetrically dividing cells.** (A) $\Lambda_P = \lambda$ exactly in the absence of noise in $\lambda$ for cells that display size control, regardless of time-additive noise in generation times or division asymmetry. Coarse-grained noise in $\lambda$ decreases $\Lambda_P$. Plot is shown for adder cells with $\alpha = 1/2$. (B) Comparison of Eq 5 with simulations for a sizer model ($\alpha = 1$) shows good agreement for small $\sigma_\lambda$, with $\sigma_t = 0$. (C) $\Lambda_P$ plotted for a range of size control strategies $\alpha$ with fixed $\sigma_\lambda$. Size control strategies weaker than an adder do not display any benefit in $\Lambda_P$ from dividing asymmetrically. $\sigma_\lambda/\lambda_0 = 0.2$, $\sigma_t = 0$. (D) Deviation from a sizer causes a relative decrease in $\Lambda_P$ for large $x$. The difference between Eqs 43 in S1 Appendix for $\Lambda_P(\alpha)$ and 5 for $\Lambda_P(\alpha = 1)$ is plotted against $x$ for deviations from $\alpha = 1$. Parameters are listed in the figure legend, with $\sigma_\lambda/\lambda_0 = 0.2$, and $\sigma_t = 0$. Data points correspond to simulations, while lines represent theoretical predictions. Error bars in (A-C) show the standard error of the mean.

asymmetrically, the growth rate gain associated with increasing $x$ shown in Fig 2C is reduced as the size control strategy weakens from $\alpha = 1$ to $\alpha = 0.6$. An adder size control strategy with $\alpha = 1/2$ shows a distinct behavior, whereby increasing $x$ has a slight tendency to *decrease* $\Lambda_P$, and this slight decrease in $\Lambda_P$ with increasing $x$ remains consistent for size control strategies weaker than an adder. This strong dependence of $\Lambda_P$ on the strategy of size control has not been observed previously in studies focusing on symmetric division, and to our knowledge is

the first instance in which $\Lambda_P$ depends on the strategy of size control in exponentially growing cells.

By expanding around $\alpha = 1$ we obtained an approximate expression for the growth rate $\Lambda_P(\alpha)$ for small $|\alpha - 1|$ (see Section 4 and Eq 43 in S1 Appendix for details). Fig 2D shows our predictions for the growth rate difference between a sizer model and cells with size control set by $\alpha$. We observe good agreement with simulations for small $x$, supporting our result that asymmetrically dividing cells with $\alpha < 1$ will have a lower $\Lambda_P$ relative to the $\alpha = 1$ sizer case.

For completeness we also explored the behavior of the population asymmetry factor $m$, showing that in the case of a sizer model $m = x$ exactly, independent of $\sigma_\lambda$ (Section 3 in S1 Appendix and S2 Fig), and that weaker strategies of size control show a weaker dependence of $m$ on $x$.

## 2.4 Generation time correlations

One recent study observed positively correlated generation times in closely related budding yeast cells [13]. When these correlations were introduced in simulations of growing populations of cells that did not regulate their size, they led to an enhancement of the population growth rate $\Lambda_P$. This prompted the authors to conclude that the epigenetic inheritance of generation times may enhance the population growth rate. We will show in Section 2.5 that the population growth rate may in theory be enhanced by introducing strong single cell growth rate correlations (for both symmetrically and asymmetrically dividing cells), however, this effect is distinct from the one discussed here [8]. The experimental observation of positively correlated generation times is surprising when contrasted with the negative correlations associated with cell size control in symmetrically dividing cells [6]. To investigate this, we adopted a model of cell cycle duration (Eq 7) that has been previously applied to analytically calculate the generation time correlation coefficients of cells growing with varying strategies of size control [16]:

$$\tau = (\ln(2) + \alpha \ln(\Delta/V_b))/\lambda + \eta. \tag{7}$$

We adopted this expression to assist in making analytical predictions for the generation time correlations. Here $\Delta$ is the mean cell size at birth, $\eta \sim N(0, \sigma_t^2)$ is a coarse grained I.I.D. noise in generation times [16], and $\lambda \sim N(\lambda_0, \sigma_\lambda^2)$ is the noisy single cell growth rate. Eq 7 arises from the growth policy $h(V_b) = 2V_b^{1-\alpha}\Delta^\alpha$, which agrees to first order with Eq 4 when Taylor expanded around the average newborn size $\Delta$. $\alpha = 0$ corresponds to a timer and $\alpha = 1$ corresponds to a sizer, and $\alpha = 0.5$ corresponds to first order with an adder. Using this model we obtained an approximate formula for the Pearson correlation coefficient (PCC) arising from cell size control in asymmetrically dividing cells in the case without noise in the single cell growth rate ($\sigma_\lambda = 0$) (see Section 9 in S1 Appendix for details), finding that positively correlated generation times can arise as a natural consequence of cell size control. This counter-intuitive result for asymmetrically dividing cells contrasts strongly with the negative generation time correlation $PCC = -\alpha/2$ that is predicted by this model for symmetrically dividing cells, but this discrepancy can be readily explained. Negatively correlated generation times arise when symmetrically dividing cells time their division events to correct for noise-induced fluctuations in cell size that were generated in previous cell cycles. For asymmetrically dividing cells there is an additional, positive term that arises due to a cell's lineage: a daughter cell that is born from a daughter cell will be smaller than the average daughter cell. In the case of an adder model, this smaller daughter cell will take a longer time to add the same volume increment $\Delta$ through exponential growth, leading to a longer than average division time. Conversely, a daughter cell that is born from a mother cell will be larger than an average daughter cell, with a

shorter than average generation time. The corresponding results hold for mother cells generated from daughter cells, and mother cells generated from mother cells. For a sizer model, all daughter cells or mother cells are born at the same average daughter or mother size, irrespective of that cell's lineage. In this case the positive term vanishes, leaving only the negative correlation arising from the correction of noise-induced fluctuations in cell size as shown in Eq 50 in S1 Appendix and S3(A) Fig.

S3 Fig shows good agreement between our predictions and the correlation coefficients measured in our simulations, both between parent cells and their progeny, and between the two cells generated from a single division event. We observe positive correlations across a broad range of $\alpha$, $x$ and $\sigma_t$ values. Using simulations we also investigated the effect of non-vanishing growth rate noise, finding that large $\sigma_\lambda$ suppressed these positive correlations, but that growth rate noise had little effect for biologically relevant regimes with $\sigma_\lambda/\lambda_0 \approx 0.15$ [4], as shown in S3E and S3F Fig). We also tested the effect of growth morphology, simulating population growth for cells dividing with a budding morphology. Doing so led to additional complexity which is not captured by our theoretical predictions for non-budding cells, and which became more pronounced for increasing $\alpha$, $\sigma_t$ and $x$ (see S4 Fig). These deviations are expected, since the effects of budding on cell division and cell cycle timing are only expected to arise when cells both regulate their size and display variability in cell size (due here to the introduction of division time noise). However, even in the case of budding cells we still find positive generation time correlations across a broad region of parameter space. The authors in Ref. [13] quote a characteristic value of $R^2 = 0.25$ for the generation time correlation between the mother and daughter cells generated from a cell division event. This aligns well with our our model's predictions, as shown in S4 Fig. Our results therefore motivate the hypothesis that epigenetically inherited division times in budding yeast may arise as a simple consequence of cell size control, without directly affecting the population growth rate as was previously thought. We again emphasize that although correlated generation times are often a consequence of cell size control, the observation of these correlations in simulated populations of cells is not in itself *sufficient* to generate cell size control.

## 2.5 Growth rate penalty and correlated growth rates

Our findings in Section 2.3 indicate that subject to our model's assumptions, a single-celled organism is expected to experience a selective pressure to minimize noise in the single cell growth rate. However, for a fixed $\sigma_\lambda$ our model predicts that an organism with strong size control might ameliorate its growth rate deficit by dividing asymmetrically. This surprising finding appears inconsistent with the observation of symmetrically dividing cells displaying both size control and noise in their volume growth rates [4], and prompted us to revisit the assumptions underpinning our modeling approach.

To first ensure that our results were robust to minor differences in model structure, we simulated cells following a more detailed inhibitor dilution model. In this model, a stable molecular inhibitor of cell cycle progression must be diluted through growth in order for cells to pass through an essential cell cycle checkpoint (known as Start in the case of budding yeast). Additional inhibitor molecules are then synthesized in the cell cycle period prior to cell division. Our prior work investigated the adder correlations that arise within an inhibitor dilution model [20]. Here we use a tunable parameter $a$ to describe the degradation of some fraction of a cell's stock of inhibitor once the cell has passed through Start. By tuning $a$ between 0 and 1, this can vary the strategy of size control between a sizer at $a = 1$ in which all a cell's inhibitor is completely degraded and newly synthesized with each cell cycle, and an adder at $a = 0$ in which no degradation takes place but inhibitor synthesis still occurs (see Section 5 in S1 Appendix for

details). The sizer case of this model with $a = 1$ shows good agreement with Eq 5 (see S1(C) Fig), while the qualitative behavior of decreasing growth rate for weaker size control strategies reproduces that obtained using Eq 4.

We also tested whether this inhibitor dilution model was able to generate robust positive correlations between the generation times of closely related cells, finding that despite quantitative differences arising from differences in model structure, the qualitative findings of Section 2.4 remain intact in this case, as shown in S4E and S4F Fig).

Fig 2D validated our use of Eq 3 for non-IGT cases of cell growth. As a further confirmation of our approach for non-IGT size control strategies, we numerically solved Eq 3 for $\Lambda_P$ based on the distribution $f_0(\tau)$ generated by our simulations (Section 6 in S1 Appendix), and compared our results to the direct fitting of $\Lambda_P$ based on the population growth over time. These results are plotted in S5 Fig and show strong agreement between these two approaches.

**2.5.1 Non-exponential growth.** Experimental evidence demonstrates that excessively large budding yeast cells ($\geq 200 fL$, relative to a population average size of $\approx 50 fL$) are known to deviate from exponential growth [23, 24]. This observation has also been predicted on theoretical grounds, due to a low DNA concentration becoming rate-limiting for transcription in excessively large cells [25]. Similarly, excessively small cells are also expected to suffer a fitness cost in their growth rate (for example, due to a limiting abundance of resources for essential cell functions). Motivated by these results, we explored the impact on the population growth rate of a growth rate penalty for cells whose volume deviates from some "optimal" value, both computationally and theoretically (see Section 7 in S1 Appendix for details). Within a biologically relevant range for $x$, S6 Fig shows a significant decrease in $\Lambda_P$ with increasing division asymmetry $x$ for cells with weak size control strategies for the parameters tested here. This result is intuitive since broad size distributions will be more penalized by a given growth rate penalty. This finding therefore highlights the need for further experiments that investigate the connection between average cell size and population growth rate, in order to place constraints on the magnitude of such a growth penalty.

**2.5.2 Correlated growth rates.** To investigate the effect of correlated single-cell growth rates $\lambda$, we used a model in which the Pearson correlation coefficient (PCC) in $\lambda$ between a parent cell and its progeny could be varied systematically [6] (see Section 8 in S1 Appendix for details). We tuned the PCC between 0 and 1, finding two qualitatively different regimes for the behavior of $\Lambda_P$. S7 Fig demonstrates that for a PCC below 0.5, the effect of growth-rate correlations is minimal, with similar qualitative behavior to that presented in Fig 2. In contrast, large growth rate correlations $\geq 0.5$ alter the effect of growth rate noise on $\Lambda_P$, leading to an *increase* in $\Lambda_P$ with increasing $\sigma_\lambda$, consistent with previous results [8]. S7 Fig further shows that within this regime of strong correlations, increasing division asymmetry *negatively* affects the population growth rate. Experimental observations in *E. coli* show weak correlations in the single cell growth rate with a PCC between mother and daughter growth rates of less than 0.1, indicating that the results of Fig 2 are expected to hold in this case [6, 26].

Our findings show that biologically relevant growth rate correlations are unlikely to be large enough to significantly alter the growth rate gains associated with asymmetric division. Further, S6 Fig only showed a substantial effect of a growth rate penalty on cells with weak strategies of cell size control. This leaves open the question of why symmetrically dividing cells are so prolific, when dividing asymmetrically with a strong size control strategy can in theory enhance the population growth rate. This discrepancy between our predictions and experimental observations may arise from the fact that the exponential growth rate is not the only physiological variable that is likely to be evolutionarily selected for. Alternatively, it may indicate that cells are unable to tune their division asymmetry without causing other adverse

physiological effects, preventing them from exploiting the growth rate gains associated with asymmetric division. We expand on these points further in the discussion.

## 3 Discussion

We study the population growth rate $\Lambda_P$ of asymmetrically dividing cells, obtaining analytic expressions for $\Lambda_P$ which were confirmed by comparison with simulations. We find that the population growth rate for cells that regulate their size is primarily determined by the single cell growth rate $\lambda$, and demonstrate that stochasticity in $\lambda$ decreases $\Lambda_P$ for a model in which noise is coarse-grained over the full cell cycle. This finding is consistent with recent work on this subject in the context of symmetric cell division [6], but conflicts with the interpretation of other studies which predicted that increased noise in generation times will enhance $\Lambda_P$, based on models that do not incorporate cell size control [7, 13]. One study presented analytical arguments to support the conclusion that the population doubling time ($T_D$) is consistently lower than the average single cell doubling time: $(\langle t_d \rangle - T_D)/T_D \geq 0$ [7]. Indeed, this inequality is still expected to hold in the case of an asymmetrically dividing population. This may readily be seen by simple application of Jensen's inequality to the average of the convex function $\langle e^{-\Lambda_P t} \rangle$ within the Euler-Lotka equation. However, within the class of models we study, the observation that the population doubling time is smaller than the average single cell doubling time does not imply that stochasticity enhances the population growth rate.

Our model further predicts that cells with strong cell size regulation can offset the growth rate deficit that noise in $\lambda$ generates by dividing asymmetrically. To our knowledge, this is the first model in which exponentially growing cells display a population growth rate that depends on the strategy of size control. Ideally, the predictions we have made here would be tested experimentally by directly varying $x$ for cells with strong size regulation and testing the population growth rate.

To reconcile our results with the abundance of symmetrically dividing organisms throughout nature, we point out that there are many possible scenarios regarding the strength of selection for a higher population growth rate. In one scenario, rapid population growth is the most strongly selected parameter in evolution over many microbial lifecycles, in which case we must conclude that some biologically relevant feature is not incorporated in our model since asymmetric division is clearly not as prolific as would be expected. In another scenario, some organisms, such as yeasts, are subject to occasional strong selection for rapid growth which may lead to asymmetric division based on the predictions we have made here, while for most organisms selection for rapid growth is less important compared to other selections such as survival in harsh environments. In this second scenario, there may be tradeoff costs to asymmetric division that are not evident in exponential growth, causing symmetric division to be favored. Indeed, recent work has demonstrated the existence of a universal tradeoff between the population growth rate and the lag time in bacteria, emphasizing that the population growth rate is not the sole parameter under selection in a given growth medium [27].

Within the first scenario, it may be the case that asymmetric division is difficult to achieve without compromising other aspects of bacterial growth (e.g. by increasing noise in the single cell growth rate), thereby preventing symmetrically dividing organisms from taking advantage of the associated growth rate gains. Another possibility is that the underlying assumptions in our model are flawed. One such assumption was that single-cell growth is truly exponential and independent of cell size. A modified version of our model included a growth rate penalty for excessively large or small cells. When we increased the size of this penalty it eliminated the growth-rate gains associated with asymmetric division, with weaker size control strategies experiencing a more severe growth rate penalty. This result highlights the importance of

measuring the variation in growth rate as cell volume deviates from the population average, to further our understanding of the potential advantages and disadvantages of different size control strategies in constraining the spread in cell size. Interest in this area has risen in recent years due to the widespread observations of adder size control strategies in a range of organisms [4, 5, 15], and recent experimental work has made significant steps towards this goal in budding yeast [24], but more work is needed.

One clear modeling prediction presented herein is that for cells with strong size control, increasing the division asymmetry while not otherwise perturbing cell physiology should cause the population growth rate to increase. Experimental tests for this could include perturbing the proteins responsible for establishing the cell midpoint as the point of contractile ring formation in rod-shaped cells, such as by perturbing Pom1 function in the fission yeast *Schizosaccharomyces pombe* [28] or the Min system in *E. coli* [29]. The prediction in either case would be that deviations from symmetric division will increase the population growth rate, provided the single cell growth rate distributions remain unchanged. Our work also highlights a clear need to assess the effect of cell size perturbations on cell growth rate. Experimental work in this area would assess whether a growth rate penalty for extremely large or small cells may function to limit the growth-rate gains associated with asymmetric division. A related, testable hypothesis is that asymmetrically dividing cells in nature exist at a local maximum in $\Lambda_P$ resulting from a balance between the aforementioned growth rate gains of asymmetric division and a growth rate penalty for unusually sized cells (S6 Fig). If this hypothesis is true, our model makes the intuitive, experimental prediction that weakening size control substantially in asymmetrically dividing cells without adversely affecting other physiological parameters will lead to a decrease in the population growth rate. This could in principle be done by perturbing the function of molecular regulators of cell size, however, the identities of these cell size control mediators remain largely unknown, meaning that perturbations to the spread in cell size may be confounded by accompanying perturbations to the average cell size.

Other simplifying assumptions in our model may also warrant consideration. We assumed that both cell types in an asymmetrically dividing population will follow the same size control strategy, however, in budding yeast this is not the case, with mother cells displaying weaker size control than daughter cells [5]. We believe this simplification is reasonable, since experimental evidence suggests that the slope of a linear regression between volume at division and volume at birth in budding yeast mother cells is between 1.1–1.3 across 5 growth media [5]. This deviation from the slope of 1 predicted by an adder model and measured in daughter cells is unlikely to have a significant effect on the population growth rate. We can estimate the magnitude of such an effect by considering the behavior our model predicts if this weaker strategy of size control were applied to the full cell population (corresponding to roughly $\alpha \approx$ 0.35–0.45). Fig 2C demonstrates that within this regime of size control strategies weaker than an adder, there is little dependence of the relationship between P and division asymmetry on $\alpha$, indicating that implementing multiple strategies of cell size control for different cell types is unlikely to change our predictions for the biologically relevant regimes. Furthermore, our predictions for positive generation time correlations in S3(A) and S4(A) Figs, or regarding the potential effects of a growth rate penalty model in S6 Fig remain qualitatively unchanged if we expand the regimes that we consider to be biologically relevant to budding yeast to include slightly weaker strategies of size control. A further assumption is that the division rate does not become limited by essential cell cycle processes. This assumption is expected to break down once larger mother cells divide rapidly enough to be limited by the replication and segregation of chromosomes, but this regime is not explored biologically in any of the asymmetrically dividing cell types we are aware of.

We found that positive generation time correlations can be generated by cell size control in asymmetrically dividing cells, contrasting with the negative generation time correlations predicted by the same model for symmetrically dividing cells. This finding motivates a hypothesis for the origin of experimentally observed epigenetic inheritance of division times in closely related budding yeast cells [13].

Our prior work found a significant effect of cell growth geometry on the success of a cell's size control strategy, predicting that within a budding growth morphology, size control is necessarily ineffective for symmetrically dividing cells [20]. Our collective results here further highlight the importance of studying the effects of different cell growth morphologies, demonstrating that even in the context of exponentially growing cells, asymmetric division can lead to unexpected and novel results. Given the range of diverse growth morphologies that are still being discovered, this demonstrates the need for further investigation of the physiological effects that can arise from novel growth morphologies [10].

## 4 Methods

All simulations of population growth were done using custom-designed code. Our simulations used discretized timesteps to track population growth, and each condition was repeated at least 100 times to generate accurate statistical averages. Populations were seeded with an asynchronous population of 100 cells in equal numbers of cell types $D$ and $M$, then allowed to propagate for 3.5 population doubling times. Cells were then randomly selected from this population and used to re-seed a new simulation that ran for 6 population doubling times. This was done to maximize the attainment of a steady state generation time distribution. The growth of this reseeded population was then used to infer the population growth rate.

To infer $\Lambda_P$ from our simulations, one may measure the growth rate directly based on cell number, or based on total population volume. As noted in [6], these values are identical for cells that display size regulation and therefore have a constant average volume $\langle V \rangle(t) = (\Sigma_{\text{cells}} V_i(t))/N(t) = \langle V \rangle$ at steady state. Since the population volume grows continuously and is readily measured in our simulations, the volume growth rate may be more accurately calculated than the number growth rate [6]. We therefore inferred $\Lambda_P$ based on measurements of the population volume growth rate throughout this text.

Section 7 in S1 Appendix explored the behavior of a cell size-dependent average volume growth rate. To ensure a non-negative growth rate for exceptionally large or small cells that were simulated according to this growth policy, whenever a cell was generated with a negative growth rate we removed that cell from consideration.

## Supporting information

**S1 Appendix. Supplementary information and derivations.**
(PDF)

**S1 Fig.** (A) Simulated values for $\Lambda_P$ plotted against division asymmetry $x$ for an adder model ($\alpha = 0.5$) following either budding or non-budding growth morphologies. Parameters match those of Fig 2A in the main text. Generation time noise causes a substantial effect on the population growth rate in the regime of extreme division asymmetries, and we observe minor deviations between budding and non-budding growth morphologies in this regime. (B) Comparison of Eq 5 in the main text with simulations for a sizer model ($\alpha = 1$) shows good agreement for small $\sigma_\lambda$. (C) $\Lambda_P$ for cells simulated according to a tunable inhibitor dilution model discussed in Section 5 in S1 Appendix. $\Lambda_P$ shows similar dependency on $x$, $\sigma_\lambda$ and size control

strategy to that shown in Fig 2B in the main text. Plotted for $\sigma_\lambda/\lambda_0 = 0.2$, $\sigma_t = 0$.
(EPS)

**S2 Fig. The asymmetry $m = (N_D − N_M)/(N_D + N_M)$ in population size between mothers and daughters at a given point in time increases with increasing division asymmetry in a manner that depends on the strategy of size control, and is independent of noise in the growth rate $\sigma_\lambda$.** (A) $m$ plotted against asymmetry $x$ for a sizer $\alpha = 1$ agrees with $m = x$ for a range of $\sigma_\lambda$ values. $m$ plotted for $\alpha = 0.2$ shows that $m$ is independent of $\sigma_\lambda$ for all size control strategies tested. (B) $m$ plotted against asymmetry $x$ for a range of size control strategies $\alpha$ show that weaker size control strategies have a decreased value of $m$.
(EPS)

**S3 Fig. Asymmetric division generates positive correlations between closeley related cells.** (A,C,E) Correlation coefficients for the generation times of daughter cells and those of their parent cells, $PCC(n, n + 1)$, plotted against division asymmetry $x$ for a range of size control strategies and noise strengths. (B,D,F) Correlation coefficients for the generation times of the daughter and mother cells generated by a cell division event ($PCC(n_M, n_D)$) plotted against division asymmetry $x$ for a range of size control strategies and noise strengths. (A) PCC plotted for variable size control strategy $\alpha$ as shown, with $\sigma_\lambda = 0$ and $\sigma_t\lambda_0 = 0.1$. Note that for $\alpha = 1$, the PCC remains fixed below zero as predicted by Eq 50 in S1 Appendix. (C) PCC plotted for variable generation time noise $\sigma_t$ as shown, with $\sigma_\lambda = 0$ and $\alpha = 0.6$. (E) PCC plotted for variable $\sigma_\lambda$ as shown, with $\sigma_t\lambda_0 = 0.1$ and $\alpha = 0.5$. (B) PCC plotted for variable size control strategy $\alpha$ as shown, with $\sigma_\lambda = 0$ and $\sigma_t\lambda_0 = 0.1$. (D) PCC plotted for variable generation time noise $\sigma_t$ as shown, with $\sigma_\lambda = 0$ and $\alpha = 0.6$. (F) PCC plotted for variable $\sigma_\lambda$ as shown, with $\sigma_t\lambda_0 = 0.1$ and $\alpha = 1/2$. Data points correspond to simulations, while dotted lines represent theory predictions. Error bars show the standard error of the mean over 100 simulated repeats.
(EPS)

**S4 Fig. Cells that have a budding growth morphology maintain positive generation time correlations in a broad regime of parameter space, while showing significant deviations from the correlations of non-budding cells for increasing $\sigma_t$, $\alpha$ and $x$ (cf. S3 Fig).** (A,C) Correlation coefficients for the generation times of daughter cells and those of their parent cells, $PCC(n, n + 1)$, plotted against division asymmetry $x$ for a range of size control strategies and noise strengths. (B,D) Correlation coefficients for the generation times of the daughter and mother cells generated by a cell division event ($PCC(n_M, n_D)$) plotted against division asymmetry $x$ for a range of size control strategies and noise strengths. (A) PCC plotted for variable size control strategy $\alpha$ as shown, with $\sigma_\lambda = 0$ and $\sigma_t\lambda_0 = 0.1$. (C) PCC plotted for variable generation time noise $\sigma_t$ as shown, with $\sigma_\lambda = 0$ and $\alpha = 0.6$. (B) PCC plotted for variable size control strategy $\alpha$ as shown, with $\sigma_\lambda = 0$ and $\sigma_t\lambda_0 = 0.1$. (D) PCC plotted for variable generation time noise $\sigma_t$ as shown, with $\sigma_\lambda = 0$ and $\alpha = 0.6$. Data points correspond to simulations, while dotted lines represent theory predictions. Error bars show the standard error of the mean over 100 simulated repeats. Cells were simulated to grow with a budding growth morphology. (E-F) Correlation coefficients for the generation times of cells simulated using an inhibitor dilution model for cell cycle progression, and dividing with a non-budding morphology. (E) Correlation coefficients for the generation times of daughter cells and those of their parent cells, $PCC(n, n + 1)$, plotted against division asymmetry $x$ for a range of size control strategies ranging between a sizer for $a = 1.0$ and an adder for $a = 0.0$. $\sigma_\lambda = 0$, and $\sigma_t\lambda_0 = 0.1$. (F) Correlation coefficients for the generation times of daughter and mother cells generated by a cell division event ($PCC(n_M, n_D)$) plotted against division asymmetry $x$ for a range of size control strategies ranging between

a sizer for $a = 1.0$ and an adder for $a = 0.0$. $\sigma_\lambda = 0$, and $\sigma_t \lambda_0 = 0.1$.
(EPS)

**S5 Fig. Comparison of simulated values for $\Lambda_P$ vs. numerically estimated values using Eq 10 in S1 Appendix.** across a range of parameter values, as described in Section 6 in S1 Appendix. Simulated growth rates based on a fit to the exponential growth of the cell population are shown as data points with error bars, while the growth rates calculated from numerical integration of the Euler-Lotka equation are shown as transparent coloured lines. Error bars show standard error of the mean. The simulations show strong agreement with the prediction of Eq 10 in S1 Appendix throughout. Results were obtained with $\sigma_\lambda/\lambda_0 = 0.2$, $\sigma_t = 0$. The dotted line corresponds to the theoretical prediction of Eq 5 in the main text for a sizer model with the same noise strength.
(EPS)

**S6 Fig. Plots of the population growth rate $\Lambda_P$ for cells undergoing asymmetric division and growing with a growth rate penalty described by Eq 45 in S1 Appendix, given $f(x) = 1 - \epsilon x^n$.** Introducing a growth rate penalty generates local maxima in $\Lambda_P$ at $x = 0$ for $n = 2$, and at finite $x$ for $n = 4$. (A-B) Predicted $\Lambda_P$ for cells growing according to a sizer size control strategy with varying penalty strength $\epsilon$ show good agreement with the predictions of Eq 47 in S1 Appendix. Simulations are generated for $\sigma_\lambda/\lambda_0 = 0.1$ and $\sigma_t = 0.0$. (C-D) Simulated predictions for cells growing with variable size control strategies for $\epsilon = 0.04$, $\sigma_\lambda/\lambda_0 = 0.1$ and $\sigma_t = 0.0$. The growth rate penalty becomes more pronounced for asymmetrically dividing cells with weaker size control strategies. Results are plotted for variable $x$, with (A, C) $n = 2$ and (B, D) $n = 4$. Data points correspond to simulations, while dotted lines represent theory predictions. Error bars show the standard error of the mean over 100 simulated repeats.
(EPS)

**S7 Fig. Plots of the population growth rate $\Lambda_P$ for cells with growth rate correlations implemented according to Eq 48 in S1 Appendix.** Growth rate correlations have strength determined by the parameter $a$. Plots show results for cells simulated to follow (A) a sizer size control strategy, or (B) an adder size control strategy. Error bars show the standard error of the mean over 100 simulated repeats.
(EPS)

## Acknowledgments

The authors would like to thank Ethan Levien and Jie Lin for helpful discussions and observations, and MCB Graphics for their help with figure illustrations.

## Author Contributions

**Conceptualization:** Felix Barber, Jiseon Min, Andrew W. Murray, Ariel Amir.

**Formal analysis:** Felix Barber, Jiseon Min, Ariel Amir.

**Funding acquisition:** Andrew W. Murray, Ariel Amir.

**Investigation:** Felix Barber, Jiseon Min, Andrew W. Murray, Ariel Amir.

**Supervision:** Andrew W. Murray, Ariel Amir.

**Visualization:** Felix Barber, Jiseon Min.

**Writing – original draft:** Felix Barber, Jiseon Min, Andrew W. Murray, Ariel Amir.

**Writing – review & editing:** Felix Barber, Jiseon Min, Andrew W. Murray, Ariel Amir.

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
