## [Decision Letter · Decision Letter 0]

6 Mar 2021

Dear Associate Professor Amir,

Thank you very much for submitting your manuscript "Modeling the impact of single-cell stochasticity and size control on the population growth rate in asymmetrically dividing cells" for consideration at PLOS Computational Biology. As with all papers reviewed by the journal, your manuscript was reviewed by members of the editorial board and by several independent reviewers. The reviewers appreciated the attention to an important topic. Based on the reviews, we are likely to accept this manuscript for publication, providing that you modify the manuscript according to the review recommendations.

Sincerely,

Attila Csikász-Nagy

Associate Editor

PLOS Computational Biology

Douglas Lauffenburger

Deputy Editor

PLOS Computational Biology

[LINK]

Reviewer's Responses to Questions

**Comments to the Authors:**

Reviewer #1: See attachment

Reviewer #2: This is a comprehensive theoretical study linking the population growth rate to size control strategies and the various single cell level stochasticity in asymmetrically dividing microbial cells, extending their study on symmetrically dividing ones. The results are valuable and insightful, and can hopefully motivate further studies on some real examples (such as the budding yeast). I have only a few minor points:

(1) The author used h(v_b) in the text (e.g. line 98) but f (v_b) in Figure 1.

(2) Line 347: both both.

(3) In several places in the text, the author used the budding yeast as examples. But their study, as they pointed out in line 111-118, is essentially entirely on the non-budding case (Figure 1B) (BTW, it would help to list some real examples of the non-budding case.) This may cause some confusion.

Reviewer #3: This article studies the impact of size control mechanism and the single-cell randomness on the population growth rate in asymmetrically dividing cells. This work is closely related to the previous work (reference [2]) by the correspondance author. The major difference is that [2] was based on symmetrically dividing cells while this manuscript is about asymmetrically dividing cells. One important result from this work is the conclusion that noise in the single-cell growth rate enhance the population growth rate for cells with strong size control. This is different from the case in symmetrically dividing cells, in which the noise in single-cell growth rate decrease the population growth.

Overall, this manuscript presented a fairly complete work with solid mathematical analysis and computer simulation based on simple models of cell size control. Both the results and the discussion are very interesting. One drawback of this work probably is the lack of direct support of wet-lab experiments. Although there were experimental results that the authors’ analysis could match with, they are not direct support. There are mutant cells that are viable in certain medium even without size control. I think the corresponding correlation data can be used to verify the validity of the analysis. I suggest that the authors make a few prediction based on possible experimental setting, so that the community could proceed and verify (or deny) the results presented with this analysis.

The other problem is related to the size control mechanism used in the analysis. I’m particularly concerned with the simple model of size control, as it is the basis of the whole analysis but it lacks of necessary discussion. For the budding yeast cells, the size control applies mainly to the G1 phase. Wet lab experiments show a clear negative correlation between the birth volume and time in the G1 phase for daughter cells with smaller sizes, but the corresponding correlation for mother cells and daughter cells with larger sizes are much lower. In the manuscript the authors are aware of the difference of cell size control mechanism for mother and daughter cells, as it was discussed a little bit in the Discussion section. I wonder how the correlation for the small volume and the time in G1 phase, rather than a whole cell cycle time is represented here. If the cell size control mechanism cannot be simply represented by equation (4) on page 5, then will all the analysis afterwards apply? Or what kind of changes will it lead to? I think this is a fundamental question this manuscript did not answer and it definitely worth some analysis and discussion.

I think these two questions I raised here are challenging and may be hard for the authors to provide immediate and complete answer. I expect the authors to give sufficient discussion on them, though.

**Have all data underlying the figures and results presented in the manuscript been provided?**

Reviewer #1: Yes

Reviewer #2: Yes

Reviewer #3: Yes

PLOS authors have the option to publish the peer review history of their article (what does this mean?). If published, this will include your full peer review and any attached files.

Reviewer #1: No

Reviewer #2: No

Reviewer #3: No

Figure Files:

Data Requirements:

Reproducibility:

References:

---

## [Decision Letter · Decision Letter 1]

14 May 2021

Dear Associate Professor Amir,

We are pleased to inform you that your manuscript 'Modeling the impact of single-cell stochasticity and size control on the population growth rate in asymmetrically dividing cells' has been provisionally accepted for publication in PLOS Computational Biology.

Best regards,

Attila Csikász-Nagy

Associate Editor

PLOS Computational Biology

Douglas Lauffenburger

Deputy Editor

PLOS Computational Biology

Reviewer's Responses to Questions

**Comments to the Authors:**

Reviewer #1: The authors have thoughtfully and carefully answered the points of my previous report. They have in particular reorganized some sections as I suggested and I think that the revised text reads very well now. I am also pleased with the way they have handled the question of the prolific symmetrically dividing cells in the discussion section. Congratulations for this nice piece of work, for me the paper is ready to be published.

**Have the authors made all data and (if applicable) computational code underlying the findings in their manuscript fully available?**

Reviewer #1: Yes

PLOS authors have the option to publish the peer review history of their article (what does this mean?). If published, this will include your full peer review and any attached files.

Reviewer #1: No

---

## [Editor Report · Acceptance letter]

16 Jun 2021

PCOMPBIOL-D-20-02158R1 

Modeling the impact of single-cell stochasticity and size control on the population growth rate in asymmetrically dividing cells

Dear Dr Amir,

I am pleased to inform you that your manuscript has been formally accepted for publication in PLOS Computational Biology. Your manuscript is now with our production department and you will be notified of the publication date in due course.

With kind regards,

Agota Szep
